# Study on the Technology of Monodisperse Droplets by a High-Throughput and Instant-Mixing Droplet Microfluidic System

**DOI:** 10.3390/ma14051263

**Published:** 2021-03-07

**Authors:** Rui Xu, Shijiao Zhao, Lei Nie, Changsheng Deng, Shaochang Hao, Xingyu Zhao, Jianjun Li, Bing Liu, Jingtao Ma

**Affiliations:** State Key Laboratory of New Ceramics and Fine Processing, Institute of Nuclear and New Energy Technology, Tsinghua University, Beijing 100084, China; xur17@mails.tsinghua.edu.cn (R.X.); zhaosj16@mails.tsinghua.edu.cn (S.Z.); niel18@mails.tsinghua.edu.cn (L.N.); changsheng@mail.tsinghua.edu.cn (C.D.); haosc@tsinghua.edu.cn (S.H.); zhaoxingyu@tsinghua.edu.cn (X.Z.); leejj@mail.tsinghua.edu.cn (J.L.); bingliu@mail.tsinghua.edu.cn (B.L.)

**Keywords:** microfluidic, high-throughput, micromixing, monodisperse droplets, internal gelation process

## Abstract

In this study, we report a novel high-throughput and instant-mixing droplet microfluidic system that can prepare uniformly mixed monodisperse droplets at a flow rate of mL/min designed for rapid mixing between multiple solutions and the preparation of micro-/nanoparticles. The system is composed of a magneton micromixer and a T-junction microfluidic device. The magneton micromixer rapidly mixes multiple solutions uniformly through the rotation of the magneton, and the mixed solution is sheared into monodisperse droplets by the silicone oil in the T-junction microfluidic device. The optimal conditions of the preparation of monodisperse droplets for the system have been found and factors affecting droplet size are analyzed for correlation; for example, the structure of the T-junction microfluidic device, the rotation speed of the magneton, etc. At the same time, through the uniformity of the color of the mixed solution, the mixing performance of the system is quantitatively evaluated. Compared with mainstream micromixers on the market, the system has the best mixing performance. Finally, we used the system to simulate the internal gelation broth preparation of zirconium broth and uranium broth. The results show that the system is expected to realize the preparation of ceramic microspheres at room temperature without cooling by the internal gelation process.

## 1. Introduction

Mixing is a necessary process for reactants to come into contact with each other before a reaction. A micromixer has the advantages of fast and uniform mixing, no contamination of the reagents and the reduction of reagent consumption [1,2], better heat and mass conduction, and can effectively realize chemical reactions sensitive to air and humidity [3] and the safer synthesis of dangerous compounds [4]. These advantages have attracted strong interest from researchers, leading to the widespread study of micromixers in DNA hybridization [5], cell activation [6,7], enzyme reactions [8], protein folding [9], water quality monitoring [10], flow chemistry [11] and the synthesis of micro-/nanoparticles, etc. [12,13,14,15].

Su, for example, in the synthesis of micro-/nanoparticles, used a T-junction mixer to mix two solutions to prepare 15–100 nm BaSO_4_ nanoparticles [16]. Wang mixed the reagent and zirconium broth in a glass capillary, and silicone oil simultaneously sheared the two solutions to form droplets and prepared 100 μm ZrO_2_ microspheres under the condition of zirconium broth flow rate of 1 μL/min [17]. Frenz embedded an external electrode on both sides of the microchannel to induce the fusion of two different component droplets through alternating current and prepared Fe_3_O_4_ nanoparticles smaller than 15 nm under the condition of an aqueous phase flow rate of 120 μL/h [18]. Zhang prepared a 1.8 mm wide and 100 mm thick micromagnetic gyromixer to achieve uniform mixing [19]. The micromixers adopted by the above researchers can be divided into passive micromixers and active micromixers according to whether there is an external power source. Passive micromixers mainly improve the mixing performance by increasing the contact area between fluids and constructing chaotic convection through the microchannels. Passive micromixers have a drawback, i.e., the mixing performance is not ideal when the Reynolds number is low, which limits their application. However, active micromixers do not have this drawback because they actively enhance mixing performance by using some form of external energy to generate chaotic convection, such as electric or magnetic fields. Active micromixers, for example micromagnetic gyromixers, also have limited application due to their complicated manufacturing process and high cost [20,21,22]. Moreover, by reducing the size of the droplets to shorten the distance of solute diffusion, researchers have currently achieved micromixing at a flow rate of μL/min or μL/h, which is unfavorable for large-scale industrial applications of micromixing. In addition, these micromixers usually have microchannels etched on polydimethylsiloxane and then thermally bonded together. If the two solutions react when mixed in the micromixer they may generate insoluble substances; for example, FeCl_3_ solution and NaOH solution generate Fe(OH)_3_ particles, which can easily block the microchannels and cause damage to the micromixer. However, a reusable, low-cost, high-throughput micromixer has not been developed yet. Therefore, there is an urgent need for a high-throughput micromixer with good mixing performance, which needs to be disassembled to clean the insoluble matter in the micromixer, so as to realize the reuse of the micromixer.

In this work, a novel high-throughput and instant-mixing droplet microfluidic system (noted as DMS) is constructed. The DMS is composed of an active magneton micromixer and a T-junction microfluidic device. Liquid droplets can be produced with such a device. The DMS can be used for the rapid and uniform mixing of two solutions, and can also be used for the preparation of micro/nano ceramic particles. It is easy to disassemble the DMS and clear the insoluble matter from the microchannel. By adjusting the structure of the T-junction microfluidic device and process parameters such as the magneton speed, and the content of surfactant, etc., the most suitable conditions for the DMS are found. The DMS is compared with the mainstream micromixers on the market and their mixing performance and the uniformity of the droplets’ sizes are analyzed. The effectiveness of the DMS is evaluated by using dispersed phases of different viscosities to simulate the preparation of zirconium broth and uranium broth in the internal gelation process.

## 2. Materials and Methods

### 2.1. Construction of High-Throughput and Instant-Mixing Droplet Microfluidic System

The high-throughput and instant-mixing droplet microfluidic system is schematically shown in Figure 1. The DMS is mainly composed of a magneton micromixer and a coaxial T-junction (1/4-28UNF, Runze Fluid) microfluidic device. A charge coupled device camera (CCD, Olympus, Tokyo, Japan) is used to monitor the dropping processs in situ. Two miscible aqueous phases are propelled into the chamber equipped with magnets at a flow rate of mL/min through the injection path by two syringe pumps (XFP12-BD, Zhongxinqiheng, China). The mixing performance of the two aqueous phases is adjusted by controlling the magneton speed. In order to facilitate the flexible rotation of the magneton in the chamber, the shape of the chamber is set to a cylinder with an inner diameter of 8 mm and a height of 6 mm. The volume of the liquid filling the chamber can be estimated to be 0.256 mL by subtracting the volume of the magneton from the volume of the chamber. When the flow rate of the two aqueous phases is 0.25 mL/min and the flow time of the two aqueous phases in the microchannel is ignored, it takes only 30 s to for the fluid to fill the entire chamber. It means that it only takes 30 s for the two aqueous phases to mix thoroughly and form droplets. In addition, in order to prevent the magneton from sending the two aqueous phases without being sufficiently mixed into the sample path beforehand, this micromixer is designed with the injection path at the bottom of the chamber and the sample path at the top of the chamber. The mixed solution as the dispersed phase is sheared into droplets by the continuous phase of silicone oil in the coaxial T-junction microfluidic device and the droplets are collected in a measuring cylinder. The droplet formation process is observed under the CCD camera. The physical image of DMS and Magneton micromixer are mentioned in detail in Appendix A.

The magneton micromixer is sealed with fastening screws and gaskets, and the pipe of the coaxial T-junction microfluidic device is fixed with inverted cone joints. When the two aqueous phases are not uniformly mixed or the mixing ratio is not appropriate to produce insoluble substances, the DMS can be easily disassembled to clean the clogged part, thereby realizing the reuse of the DMS and greatly reducing the cost compared with clogged and scrapped micromixer of the previous researchers.

The DMS increases the contact area of the two aqueous phases by using the magneton rotation to generate chaotic convection, and when the mixed solution is sheared into droplets, the solute diffusion distance is shortened to improve the mixing performance. Therefore, the DMS combines the characteristics of the active and passive micromixers.

### 2.2. Mixing-Target Liquids

In order to prevent the DMS from firstly clogging, deionized water was used as the aqueous phase in the process of preparing monodisperse droplets. In the comparison of mixing performance, two portions of 150 mL deionized water were added with 1 g of pigment. According to the mixing principle of the pigment, the same amount of the sky blue aqueous phase and the lemon yellow aqueous phase will become the dispersed phase of kelly green. The mixing performance is judged based on the color uniformity of the collected pictures by the CCD camera. In addition, a certain amount of polyvinyl alcohol abbreviated as PVA (Mw 13,000–23,000, Sigma-Aldrich (Munich, Germany)) was added to water with pigment to simulate a zirconium broth and a uranium broth. In order to prevent the addition of the pigment from affecting the formation of monodisperse droplets, the density, viscosity, and interfacial tension of different aqueous phases with the appropriate continuous phase are measured, as shown in Table 1. The composition of this suitable continuous phase, which will be given in the third part, is 83.6 mPa·s silicone oil (Aladdin, Shanghai, China) with 2% *v*/*v* Dow Corning 749 (Dow Corning, Midland, TX, USA).

It can be seen from Table 1 that the viscosity of the water and water with PVA after adding the pigment will increase a few mPa·s, and the density and interfacial tension are basically unchanged compared with the original solution. When the same amount of sky blue pigment solution is mixed with the lemon yellow pigment solution to obtain the kelly green pigment solution, compared with the two solutions before mixing, the density and interfacial tension of the kelly green pigment solution are basically unchanged, and the viscosity is slightly reduced. Zirconium broth and uranium broth are nearly saturated solutions, so it is normal that the density of water with PVA and pigment is lower than that of zirconium broth and uranium broth. Unlike macroflow, the interfacial tension plays a dominant role in microfluidics and the effect of gravity is usually negligible. In other words, the density difference between the simulated solution and the broth can be ignored in the DMS.

In general, deionized water is used as the aqueous phase and combined with the DMS to prepare monodisperse droplets. This mixing performance is quantitatively characterized by analyzing the color uniformity of the kelly green pigment solution mixed from the sky blue solution and the lemon yellow solution.

### 2.3. Characterization

The density of the aqueous phases is measured by a liquid densitometer. The viscosity of the aqueous phases is measured with an LVDV-1 digital rotation viscometer (Shanghai Jingtian Electronic Instrument Co., Ltd., Shanghai, China). The picture of the droplets formed at the capillary port and the picture of the droplets of the collecting cylinder are captured by an Olympus IX71 fluorescence microscope (Olympus, Tokyo, Japan). Through image recognition, the size and coefficient of variation of the droplets in these pictures are extracted, as shown in Figure 2A. The program automatically recognizes the number and outline of the droplets in the picture (Figure 2B), and calculates the size and coefficient of variation of the droplets. The specific identification principle is mentioned in the previous article by the research group [25]. The simplified variance normalization method is used to characterize the mixing performance (noted as *MP*) [1], as shown in Equation (1) where $m_{i}$ is the gray value of the i-th point in the picture, and $\bar{m}$ is the average gray value on the picture, and n is the number of pixel points in the picture.

The larger the value of *MP*, the better the mixing performance.
(1)$$MP=1-\frac{1}{\bar{m}}\sqrt{\frac{\sum_{i=1}^{n}{\left(m_{i}-\bar{m}\right)}^{2}}{n}}$$

## 3. Results and Discussion

### 3.1. The Structure of the T-Junction Microfluidic Device

The T-junction microfluidic device is a key device for forming monodisperse droplets. According to the flow direction of the continuous phase and the dispersed phase, the T-junction microfluidic device can be divided into T-junction perpendicular flow, T-junction transverse flow and coflowing. When the flow rate of dispersed phase and the continuous phase is 1 mL/min and 4 mL/min, respectively, and the continuous phase viscosity is 66 mPa·s, the droplets formed by the three flow structures are observed. It can be seen from Figure 3 that the size of the droplets prepared by the coaxial T-junction microfluidic device is the most uniform and the coefficient of variation is less than 5%, which meets the requirements of monodisperse droplets [26].

The reason for the difference is related to the droplet formation mechanism of these three flow structures. T-junction perpendicular flow and T-junction transverse flow mainly use the pressure difference before and after the droplet to break the droplet. In T-junction perpendicular flow, the dispersed phase is squeezed into a continuous liquid column that moves in in the microchannel, resulting in uneven droplet sizes (Figure 3A). In T-junction transverse flow, the dispersed phase is squeezed into discontinuous liquid columns in the microchannel and satellite droplets are generated, resulting in uneven droplet sizes (Figure 3B). The mechanism of coflowing by the coaxial T-junction microfluidic device to generate droplets is Kelvin–Helmholtz instability. That is, when the difference between the flow rate of the two phases exceeds a certain range, monodisperse droplets will be generated, as shown in Figure 3C. In addition, due to the interfacial tension of the droplets and the continuous phase, the sphericity of the droplets formed by these three flow structures is all less than 1.05 (Figure 3D), which is good. The sphericity will not be discussed in the subsequent droplet evaluation process. Therefore, the suitable structure of the T-junction microfluidic device that generates monodisperse droplets is coflowing, that is the coaxial T-junction microfluidic device.

### 3.2. The Magneton Rotation Speed

The rotational speed of the magneton directly affects the mixing performance of the liquid in the chamber of the magneton micromixer. Equal amounts of water with lemon yellow pigment and water with sky blue pigment are pushed into the chamber to observe the mixing performance under different magneton speeds. The mixing performance at different magneton speeds can be seen from Figure 4. When the magneton speed is 0 r/min, the mixing performance reaches 0.800. This is solely due to the solute diffusion between the two pigment solutions. It can be seen from the picture of the chamber in the magnetic micromixer at 0 r/min that there are still some lemon yellow solutions that are not completely mixed. When the magneton speed increases from 0 r/min to 600 r/min, the mixing performance is significantly improved, reaching 0.944. However, when the magneton speed further increases from 600 r/min to 1200 r/min, the improvement of mixing performance reaches plateau. Therefore, 600 r/min was chosen as the magneton speed of this experiment.

### 3.3. The Content of Surfactant

Dow Corning 749 (Dow Corning Co., Ltd.) is decamethyl-cyclopentasiloxane and trimethylated silica as the surfactant, which is one of the oil-soluble polymers. The surfactant is used to prevent coalescence between droplets by reducing the interfacial tension between the continuous phase and the dispersed phase. The surfactant is added to 50 mPa·s silicone oil as the continuous phase according to the volume ratio. When the flow rate of dispersed phase and the continuous phase is 0.5 mL/min and 2 mL/min, respectively, and the magneton speed is 600 r/min, the size and coalescence of droplets are observed, as shown in Figure 5. The droplets’ sizes decrease with the increase in the content of surfactant. This can be attributed to the increase in the surfactant, which reduces the interfacial tension between the dispersed phase and the continuous phase. The decreased interfacial tension is beneficial to generation of smaller droplets by the DMS. The coefficient of variation of the droplets’ sizes after the addition of surfactants does not have obvious regularities in Figure 5. It can be seen from the physical map of the collected droplets that the surfactant has little effect on the coalescence of the droplets. Then, in order to prevent the excessively high content of the surfactant from affecting the subsequent experiments of preparing monodisperse microspheres by DMS, 2% *v*/*v* of the surfactant was added to the silicone oil.

### 3.4. Factors Affecting Droplets’ Sizes and Coefficient of Variation

When 2% *v*/*v* of the surfactant is added to the silicone oil, the viscosity of the silicone oil needs to be re-measured as the viscosity of the continuous phase. When 2% *v*/*v* the surfactant is added to the silicone oil of different viscosity as the continuous phase, and the flow rate of the dispersed phase and the continuous phase is 0.5 mL/min and 2 mL/min, respectively, the magneton speed is 600 r/min, and the coaxial T-junction microfluidic device is used, the relationship between the droplet’s size and coefficient of variation and various factors is obtained, as shown in Figure 6A. It can be seen that as the viscosity of continuous phase increases, the size of the droplets decreases, and the coefficient of variation decreases from the initial 0.047 to 0.003. This is because the viscosity of the continuous phase increases, which increases its shearing force on the dispersed phase, resulting in a smaller and more uniform droplet sizes. Then, when the viscosity of the continuous phase is very large, it will increase the flow resistance of the continuous phase and the dispersed phase in the microchannel, which is not conducive to high-throughput droplet microfluidic systems. Therefore, a moderate viscosity of continuous phase (83.6 mPa·s) was selected. From the physical image of the droplets collected, which formed at 83.6 mPa·s in a measuring cylinder (Figure 6A), it can be seen that the droplets are uniform in size and have good sphericity.

The effect of the flow rate of the continuous phase is investigated under the condition that the viscosity of the continuous phase is 83.6 mPa·s, the flow rate of the dispersed phase is 0.5 mL/min, 2% *v*/*v* surfactant is used, the magneton speed is 600 r/min, and the coaxial T-junction microfluidic device is used. The results are presented in Figure 6B. With the increase in the continuous phase flow rate, the droplet’s size rapidly decreases and the coefficient of variation of the droplets fluctuates around 0.007. The volume of the continuous phase is only 100 mL on the syringe pump. Considering the initial bubble elimination time of the DMS and the time for the droplets to reach stable generation conditions, the DMS requires at least 50 min of running time to ensure accurate data collection. Therefore, 2 mL/min is chosen as the flow rate of continuous phase.

The influence of the flow rate of dispersed phase is investigated under the condition that the viscosity of the continuous phase is 83.6 mPa·s, the flow rate of the continuous phase is 2 mL/min, 2% *v*/*v* surfactant is used, the magneton speed is 600 r/min, and the coaxial T-junction microfluidic device is used. The results as shown in Figure 6C indicate that as the flow rate of the dispersed phase increases, the droplets’ sizes also increase. When the flow rate of the dispersed phase is 0.9 mL/min, the droplets’ size sin the microchannel become larger and the distance between the droplets becomes smaller, because the shear force of the continuous relative dispersed phase becomes smaller. This will easily cause the droplets of the microchannel to collide and the collected droplets in the measuring cylinder will be of uneven size. For example, when the flow rate of the dispersed phase is 0.9 mL/min, the coefficient of variation of the droplets will reach 0.04. However, if the flow rate of the droplets is small, it is not conducive to the high-throughput preparation of monodisperse droplets. Therefore, a moderate dispersed phase flow rate (0.5 mL/min) was selected.

The effect of the flow ratio of the continuous phase to the dispersed phase is analyzed under the condition that the viscosity of the continuous phase is 83.6 mPa·s, the flow rate of the dispersed phase is 0.5 mL/min, 2% *v*/*v* surfactant is used, the magneton speed is 600 r/min, and the coaxial T-junction microfluidic device is used, as shown in Figure 6D. As the flow ratio of the continuous phase to the dispersed phase becomes larger, the size of the droplets decreases rapidly, and the coefficient of variation of the droplets fluctuates at 0.01, which can be ignored. Since the inner diameter of the outlet tube of the microchannel is 1600 μm, the size of the droplets cannot be too large to avoid friction between the droplets and the tube wall of the microchannel. This means that the larger the flow ratio of the continuous phase to the dispersed phase, the better. However, considering the stable working time of the DMS, 4:1 was chosen as the best flow ratio of the continuous phase to the dispersed phase.

Therefore, using deionized water as the aqueous phase, the best conditions for the DMS to prepare monodisperse droplets are found—that is, in the coaxial T-junction microfluidic device, the magneton speed is 600 r/min, and the content of surfactant is 2% *v*/*v*, the viscosity of continuous phase is 83.6 mPa·s, the flow rate of continuous phase and dispersed phase is 2 mL/min and 0.5 mL/min, respectively, and the flow ratio of the continuous phase to the dispersed phase is 4:1.

### 3.5. Correlation Analysis of Influencing Factors

As can be seen from the above, there are many factors that affect the size of the droplets prepared by the DMS. Mathematical statistics and numerical simulations are widely used in scientific research [27,28,29]. The SPSS software (version 22.0) developed by IBM (Armonk, NY, USA) is used to do Pearson correlation analysis of these influencing factors and the droplets’ sizes. The correlation coefficient and significance of each influencing factor are shown in Table 2. It can be seen that except for the flow rate of the dispersed phase, the other influencing factors are negatively related to the droplets’ sizes. The flow ratio of the continuous phase to the dispersed phase is varied by maintaining the flow rate of the dispersed phase at a constant rate while the flow rate of the continuous phase is changed. Its correlation coefficient is basically the same as that of the flow rate of the continuous phase; both are about 0.974. In addition, the viscosity of the continuous phase has the largest correlation coefficient, followed by the flow rate of the continuous phase, then the flow rate of the dispersed phase and finally the content of the surfactant. This shows that the viscosity of the continuous phase has the closest relationship with the droplets’ sizes in the DMS, which is significant at the level of 0.01. Therefore, the droplets’ sizes can be changed mainly by changing the viscosity and flow rate of the continuous phase.

### 3.6. Mixing Performance and Uniformity of Droplets’ Sizes

In order to verify the mixing performance of the DMS, the DMS is compared with the mainstream micromixers on the market. The aqueous solutions of sky blue pigment and lemon yellow pigment (as shown in Table 1) are respectively passed into the micromixer at the dispersed phase and are sheared into droplets under the abovementioned optimal droplet generation conditions. The droplet generation drawings in the microchannel drawing are shown in Figure 7.

Figure 7A shows that the droplets prepared by DMS are evenly mixed in color, and Figure 7B shows that the droplets prepared by the T-junction microfluidic device, which is composed of a T-junction micromixer and the coaxial T-junction microfluidic device, are yellow on the outermost surface and have uniform internal color. The colors of the two pigment solutions in the droplets prepared by the serpentine micromixing chip in Figure 7C are still clearly visible. Moreover, the two pigment solutions have gone through eight u-shaped bends in the serpentine micromixing chip and they are still not evenly mixed. Their mixing performance is quantitatively extracted, as shown in Figure 8A. It can be seen from Figure 8A that the mixing performance of the collected droplets in a measuring cylinder is better than that of the generated droplets in the microchannel because the solute diffusion inside the droplets increases mixing performance when the generated droplets move into the measuring cylinder through the microchannel. Under the same flow rate of the dispersed phase, whether it is the generated droplets or the collected droplets, the mixing performance of DMS is the best, followed by the T-junction microfluidic device, and the serpentine micromixing chip comes last. This is because both T-junction microfluidic device and serpentine micromixing chip are passive micromixers, which mainly rely on solute molecular diffusion for mixing, while DMS is an active micromixer, which is prone to generating chaotic convection and has the best mixing performance among these microfluidic devices.

The magneton and serpentine microfluidic device in Figure 7D is integrated by the magneton micromixer and a serpentine micromixing chip. When the magneton speed is 600 r/min, the magneton and serpentine microfluidic device and the DMS are considered to reach the same degree of mixing. This is explained from the fact that the mixing performance of the two is similar in Figure 8A. However, the magneton and serpentine microfluidic device cannot achieve high throughput. Under the conditions of 0.5 mL/min for the dispersed phase and 2 mL/min for the continuous phase, the dispersed phase cannot be sheared into droplets by the continuous phase. The continuous phase can flow counter-currently into the chamber with the magneton that mixes the aqueous phase solutions, hindering the mixing of the solutions, resulting in the inability to form droplets. Only when the flow rate of the dispersed phase is 20 μL/min and the flow rate of the continuous phase is 100 μL/min can the droplets be formed. At a similar mixing performance level, the DMS can achieve a flow rate of mL/min to prepare droplets while the magneton and serpentine microfluidic device can only prepare droplets at μL/min, which directly illustrates the superiority of the DMS structure. It can be seen from Figure 8B that the size of the droplets generated by the DMS and the T-junction microfluidic device is very uniform, and the coefficient of variation is below 0.006. The coefficient of variation of the droplets by the serpentine micromixing chip and magneton and serpentine microfluidic device exceeded 0.5, indicating that the size distribution of the droplets prepared by these two micromixers is very large.

Therefore, compared to these three kinds of micromixers, the DMS has the best mixing performance and droplet size uniformity. In principle, the coaxial T-junction microfluidic device prepares droplets which are more stable under force on a three-dimensional flow scale than other microfluidic devices. The magneton micromixer has a simple structure and uniform mixing. It makes the flow resistance in the DMS far smaller than that of the complex microchannels in other microfluidic devices, thereby achieving the preparation of high-throughput uniformly mixed monodisperse droplets.

### 3.7. Simulated Broths Experiment by DMS

In the preparation of ZrO_2_ and UO_2_ gel microspheres by the internal gelation process, the metal ion solution needs to be mixed with a mixed solution of urea and hexamethylenetetramine (noted as HMUR solution) to form a zirconium or uranium broth, and the broth is rapidly dispersed into droplets and falls into hot silicone oil to form gel microspheres [30,31], which can be found in Appendix A. However, hexamethylenetetramine is thermally unstable and easily decomposes into ammonium hydroxide, leading to the premature precipitation and gelation of metal ions to block the dispersing device. The reaction can be simplified into Equation (2) [17,30,31] and $M^{n+}$ is the metal ion. Therefore, the metal ion solution and the HMUR solution need to be cooled and mixed at 5 °C, which is not conducive to large-scale industrial production. Moreover, when the metal ion solution and the HMUR solution are not sufficiently mixed, certain positions in the mixed solution will reach the pH at which the metal ion gelation reaction occurs, so the broth will quickly gel and block the device.
(2)$$M^{n+}+nOH^{-}\to M{\left(OH\right)}_{n}$$

The water with PVA and sky blue pigment solution is used to simulate metal ion solution and the water with PVA and lemon yellow pigment solution is used to simulate HMUR solution. The two kinds of solutions are mixed into the kelly green pigment solution similar to the interfacial tension and viscosity of the broth by the magneton micromixer, and then droplets are formed on the coaxial T-junction microfluidic device under the abovementioned optimal droplet generation conditions. The pictures of the collected droplets in the measuring cylinder are shown in Figure 9A,B. The droplet size and the coefficient of variation and the mixing performance of the DMS for these two simulated broths can be seen from Figure 9C.

It can be seen from the Figure 9A,B that the droplets of simulated zirconium broth and simulated uranium broth are formed by DMS with good sphericity and uniform size and uniform mixing. The coefficient of variation of the droplets is less than 0.01. The mixing performance is greater than 0.95 for the generated droplets in the microchannel and the collected droplets in the measuring cylinder, which can be regarded as uniformly mixed. In addition, the size of the droplets formed by the simulated uranium broth is slightly smaller than that of the simulated zirconium broth. The viscosity of the simulated uranium broth is greater than that of the simulated zirconium broth. However, previous studies have shown that changing the viscosity of the dispersed phase will not change the force state of the droplets in the microchannel. So, it is normal that the size of the droplets slightly changes with the viscosity of the dispersed phase [32].

The internal gelation process is a process of hydrolysis of metal ions which is heavily dependent on temperature. When the temperature rises from 5 °C to 20 °C, the protonation of HMTA and the decomposition of protonated HMTA will accelerate, leading to an increase in the pH of the broth and promoting the hydrolysis of the metal ions [23,24]. For example, when the temperature rises from 5 °C to 20 °C, the stability time of the zirconium broth is reduced from 5 h to 1 h, and the stability time of uranium broth is reduced from 16 h to 200 s [23,24]. Because of the short stabilization time, it is difficult to achieve a continuous internal gelation process at room temperature without cooling to prepare the zirconia or uranium oxide microspheres. The time for DMS to mix and form droplets is only 30 s, much less than 200 s and 1 h. Therefore, it is expected that the mixing process using the DMS will not strongly affect the hydrolysis and gelation process of the metal ions such that gel microspheres can be prepared at room temperature by DMS without cooling the precursor solution.

## 4. Conclusions

In this study, a novel high-throughput and instant-mixing droplet microfluidic system is designed for solution mixing and preparation of micro-/nanoparticles. The system is detachable and it is easy to clean any blockages in the microchannel, which realizes the reuse of the system and greatly reduces the cost. Moreover, the system can mix the solution uniformly and produce droplets of uniform size at a flow rate of mL/min, which overcomes the shortcomings of low droplet yield and easy clogging of the micromixing chips on the market.

The results show that the best conditions for the DMS to prepare uniform mixing and monodisperse droplets with good sphericity are in the coaxial T-junction microfluidic device, where the magneton speed is 600 r/min, and the content of surfactant is 2% *v*/*v*, the viscosity of continuous phase is 83.6 mPa·s, the flow rate of continuous phase and dispersed phase is 2 mL/min and 0.5 mL/min, respectively, and the flow ratio of continuous phase to dispersed phase is 4:1. The viscosity and flow rate of continuous phase have a major impact on monodisperse droplets of different sizes. The DMS achieves the preparation of monodisperse droplets with better mixing performance than three micromixing chips on the market. Moreover, the simulation broths are used to simulate the preparation of zirconium and uranium gel microspheres in the internal gelation process by the DMS. The DMS can potentially realize the continuous production of ZrO_2_ and UO_2_ ceramic microspheres without cooling at room temperature. Thus, the DMS is expected to meet the demands in various fields, including the high-volume industrialization of microfluidics, micromixing, and micro-/nanoparticles.

## Figures and Tables

**Figure 1 materials-14-01263-f001:**
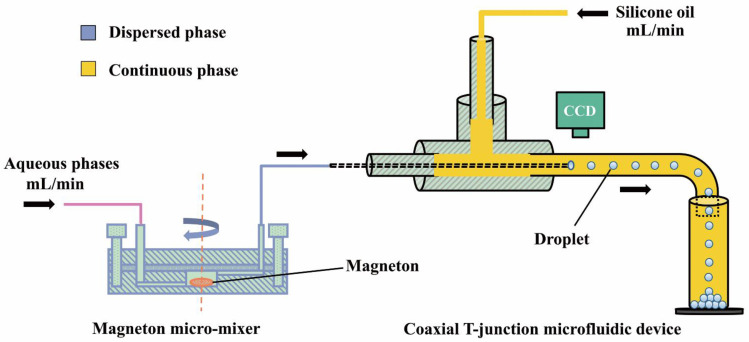
Schematic drawing of the droplet microfluidic system (DMS) for preparing monodisperse droplets.

**Figure 2 materials-14-01263-f002:**
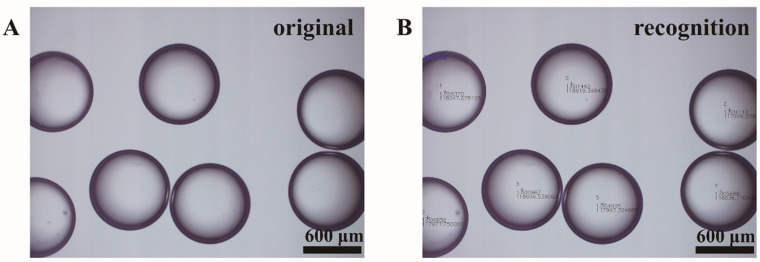
(**A**) The original image of droplets. (**B**) The image after program recognition.

**Figure 3 materials-14-01263-f003:**
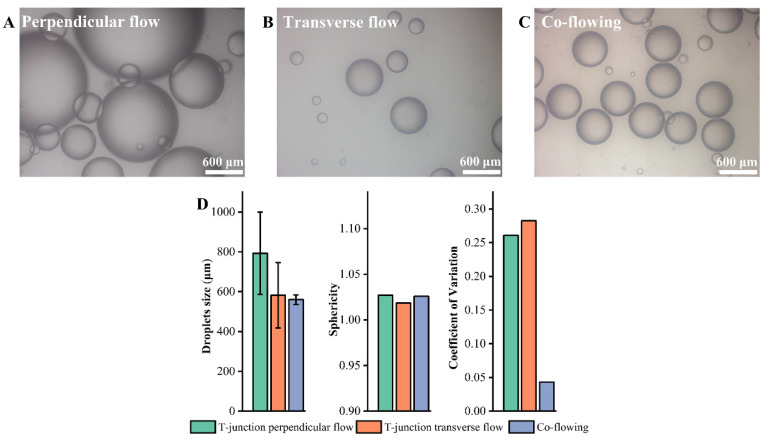
(**A**) T-junction perpendicular flow. (**B**) T-junction transverse flow. (**C**) Co-flowing. (**D**) Droplets’ sizes and sphericity and coefficient of variation by the three structures.

**Figure 4 materials-14-01263-f004:**
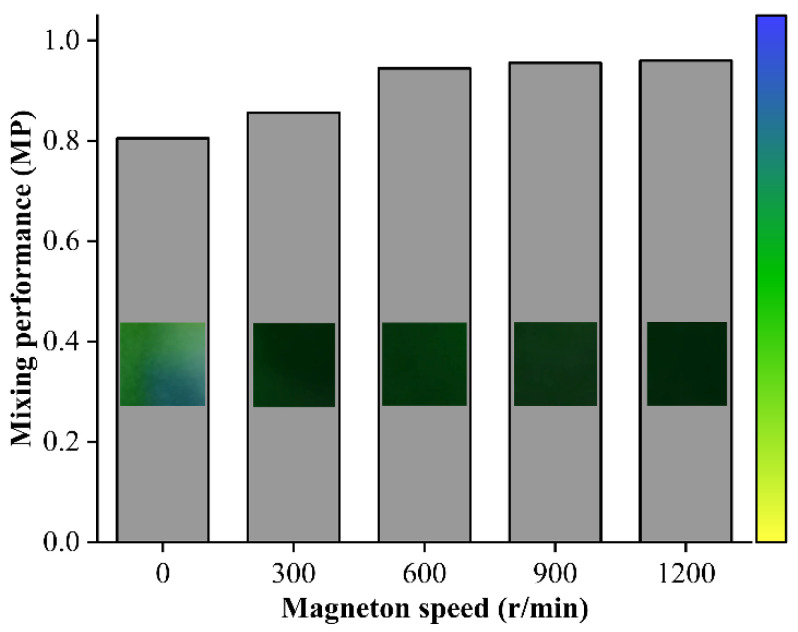
Mixing performance of different magneton speeds.

**Figure 5 materials-14-01263-f005:**
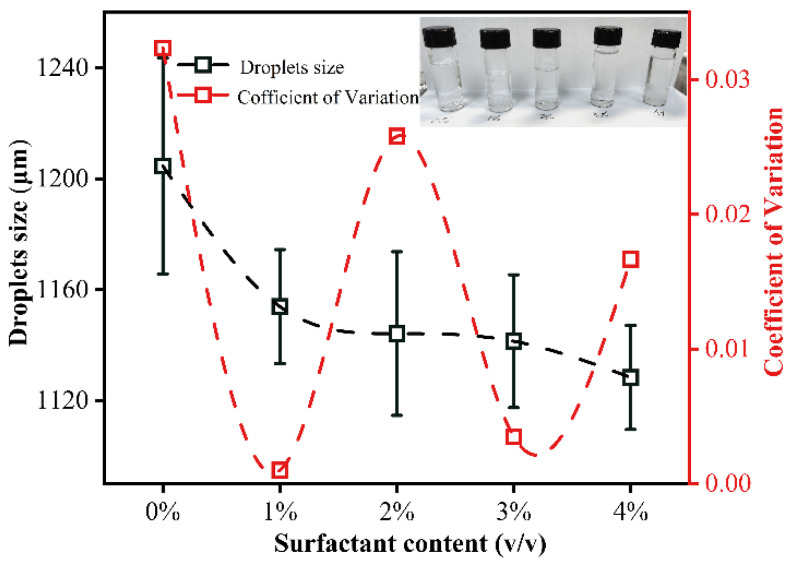
The variation of the content of surfactant with droplet’s size and coefficient of variation.

**Figure 6 materials-14-01263-f006:**
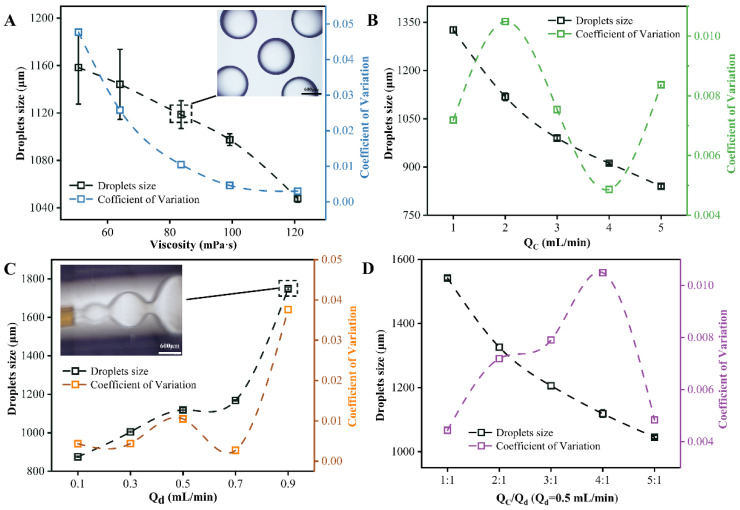
(**A**)The viscosity of continuous phase. (**B**) The flow rate of continuous phase (Q_c_). (**C**) The flow rate of dispersed phase (Q_d_). (**D**) The flow ratio of continuous phase to dispersed phase.

**Figure 7 materials-14-01263-f007:**
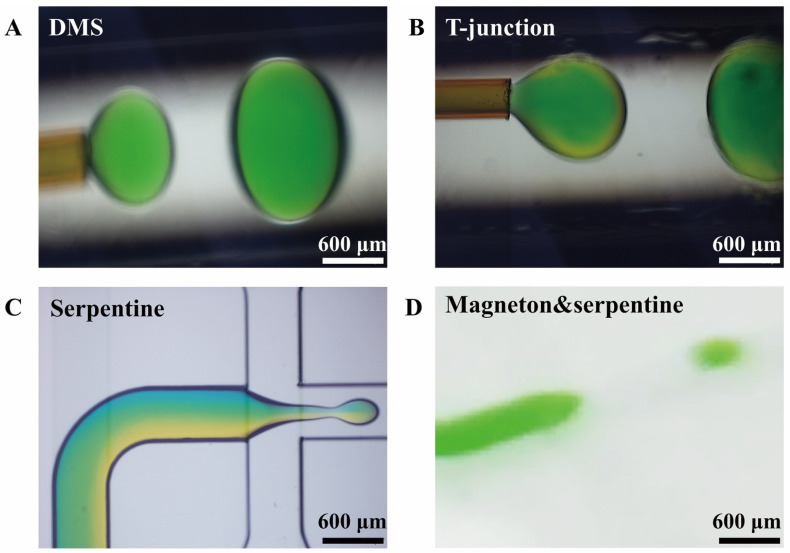
(**A**) DMS. (**B**) T-junction microfluidic device. (**C**) Serpentine micromixing chip. (**D**) Magneton and serpentine microfluidic device.

**Figure 8 materials-14-01263-f008:**
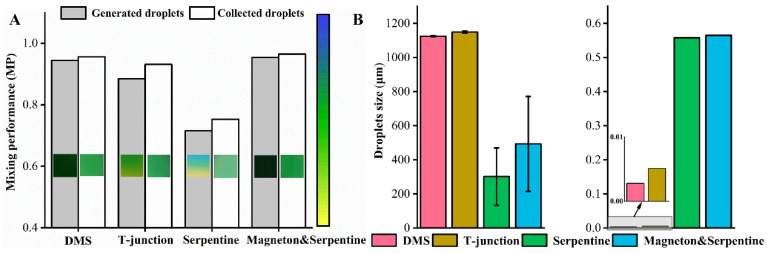
(**A**) Mixing performance of different microfluidic devices. (**B**) Droplets size and coefficient of variation of different microfluidic devices.

**Figure 9 materials-14-01263-f009:**
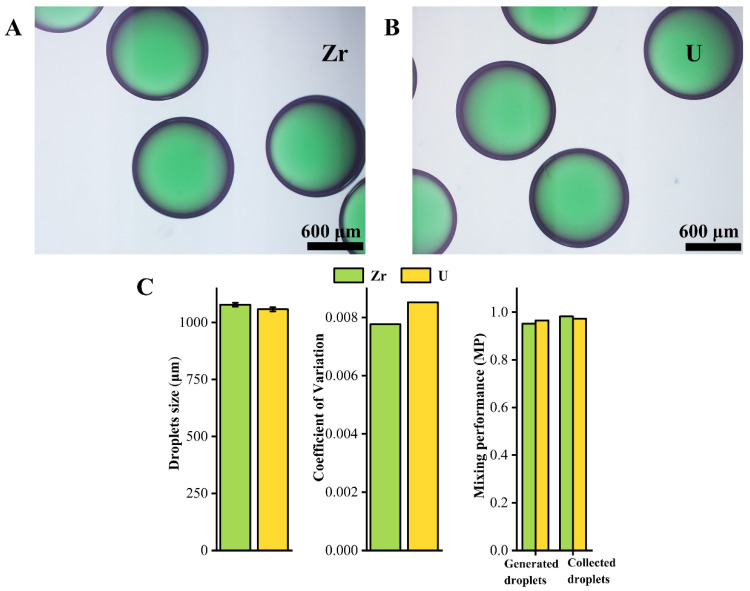
(**A**) Collected droplets of simulated zirconium broth by DMS. (**B**) Collected droplets of simulated uranium broth by DMS. (**C**) Characterization of droplets.

**Table 1 materials-14-01263-t001:** The density, viscosity, and interfacial tension of different aqueous phases.

Samples	Density (g/cm^3^)	Viscosity (mPa·s)	Interfacial Tension (mN/m)
Deionized water	1.000	1.0	20.7
Water with lemon yellow pigment	1.013	8.1	20.5
Water with kelly green pigment	1.007	7.5	20.0
Water with sky blue pigment	1.001	7.8	20.2
Zirconium broth [23]	1.211	7.0	20.9
Uranium broth [24]	1.512	14.5	21.0
Water with PVA and kelly green pigment for simulating zirconium broth	1.003	8.1	19.6
Water with PVA and kelly green pigment for simulating uranium broth	1.008	15.2	20.0

**Table 2 materials-14-01263-t002:** Pearson correlation coefficient of influencing factors and droplet size.

The Influencing Factors	Pearson Correlation with Droplets’ Size
The content of surfactant	−0.886 ^α^
The viscosity of continuous phase	−0.987 ^β^
The flow rate of continuous phase	−0.973 ^β^
The flow rate of dispersed phase	0.900 ^α^
The flow ratio of continuous phase to dispersed phase	−0.974 ^β^

^α^ Correlation is significant at the 0.05 level (2-tailed); ^β^ Correlation is significant at the 0.01 level (2-tailed).

## Data Availability

Data is contained within the article or Appendix A. The data presented in this study are available in [Study on the Technology of Monodisperse Droplets by a High-Throughput and Instant-Mixing Droplet Microfluidic System].

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
