# Peer review of "Study on the Technology of Monodisperse Droplets by a High-Throughput and Instant-Mixing Droplet Microfluidic System"

_materials, 2021, doi:10.3390/ma14051263_

Round 1

Reviewer 1 Report

In this work the authors develop and text a new approach for high throughput, of the order of mL/min, micromixing and monodisperse droplet preparation of aqueous solutions into silicone oil.  The mixing is effected in an active magneton micromixer into which the two aqueous solutions are fed.  The droplets are then developed by feeding the output of the mixer to a coaxial T-junction microfluidic device.  Two different pigments are used (yellow and blue) that allowed for a direct optical observation of the degree of mixing.  In addition, optical observations with a CCD camera allowed to evaluate the size, shape (sphericity) and size distribution of the resultant droplets as various parameters are varied like the magneton speed, the amount of surfactant, the viscosity of the continuous phase and the flowrates of the disperse and continuous phases leading to optimum design parameters.  The authors also showed the improved performance obtained with their proposed device versus 3 other alternate designs.  Finally, the authors suggest the potential suitability of their device in the development of ZrO2 and UO2 gel microspheres of usefulness in nuclear reactors by applying it to suitably adjusted (through the addition of polyvinyl alcohol) for viscosity simulant systems.

This is a carefully  performed experimental work of a potentially useful micromixing and droplet formation device that convincingly demonstrates its performance in generating uniform spherical droplets of well-mixed aqueous ingredients in silicon oil in throughputs of the order of mL/min.  However, there are a few concerns regarding the window of applications that can covered, as described in the detailed comments below, that the authors are asked to respond with in a revised version before their work can be recommended for publication.

Detailed comments

  1. The size of the produced droplets is marginally in the micron scale (more in the millimeter scale)---see, for example, Figs. 5, 6, 8. Is this a restriction of the device?  What is the window of feasible sizes that can be realistically produced (by the tested or suitably scaled device) and what are the implications to the obtained throughputs?
  2. A significant motivation for this work is its potential use for the preparation of UO2 and ZnO2 gel microspheres. However, the authors have not applied their device to the actual systems, but rather to so called “simulants” though which they simply matched the viscosities of the actual suspensions.  As those suspensions may well have a much more complicated inner structure that that of the simulants, a concern is whether the latter can be used to actually predict their mixing behavior.  The density differences are also notable.  Is there additional evidence pointing out for the suitability of the proposed simulants?  Also, in the previously quoted works (namely, reference 28 provided, as well as previous work by some of the authors that has not be mentioned here—see reference 1 below) the droplets size was much smaller of the order of 100 microns.  Is that a factor?  Related to remark (1) above, can the proposed device be adjusted to the smaller size and how is that expected to affect the throughput?  Finally, reference 1 below, suggests that external gelation leads to better quality particles as opposed to the internal gelation that is attempted to be simulated here.  Can the deficiencies detected in reference 1 associated with internal gelation production be circumvented?  A more direct comparison of the proposed device against that used in reference 1 is recommended.
  3. Some editorial remarks:
  4. a) Line 101, it is the first-time polyvinyl alcohol is mentioned, but not the acronym (PVA) that is though mentioned several times later---this needs to be added.
  5. b) In figure 3A,B and C along with the larger droplets there are much smaller ones: those are bubbles?
  6. c) Line 184: It is mentioned that the performance slows down, however, Figure 4 shows that rather it reaches a plateau.
  7. d) Line 196: The addition of surfactant is mentioned, but not what the surfactant was and the effects that its addition to the surface tension are---those are needed to better interpret the results shown in Figure 5.
  8. e) Line 466, reference 27: Please correct the name of the journal.

Reference

[1]  Guogao Wang, Jingtao Ma, Yong Gao, Xingyu Zhao, Shaochang Hao, Changsheng Deng, Bing Liu, A comparative study of small-size ceria–zirconia microspheres fabricated by external and internal gelation, J Sol-Gel Sci Technol (2016) 78:673–681.

Author Response

Outline of changes made: 
1. Section 2.2, paragraph 2 (Line 17 on the fifth page), we add the abbreviation of polyvinyl alcohol. 
2. Section 3.2, paragraph 1 (Line 8 on the tenth page), the words “slow down” is corrected to “reaches plateau”. 
3. Section 3.3, paragraph 1 (Line 14~17 on the tenth page), We add an explanation of the surfactant Dow Corning 
749. 
4. References, the journal name of Reference 27 is corrected. 

Replies to the comments of Reviewer 1:

In this work the authors develop and text a new approach for high throughput, of the order of mL/min, micromixing and monodisperse droplet preparation of aqueous solutions into silicone oil. The mixing is effected in an active magneton micromixer into which the two aqueous solutions are fed. The droplets are then developed by feeding the output of the mixer to a coaxial T-junction microfluidic device. Two different pigments are used (yellow and blue) that allowed for a direct optical observation of the degree of mixing. In addition, optical observations with a CCD camera allowed to evaluate the size, shape (sphericity) and size distribution of the resultant droplets as various parameters are varied like the magneton speed, the amount of surfactant, the viscosity of the continuous phase and the flowrates of the disperse and continuous phases leading to optimum design parameters. The authors also showed the improved performance obtained with their proposed device versus 3 other alternate designs. Finally, the authors suggest the potential suitability of their device in the development of ZrO2 and UO2 gel microspheres of usefulness in nuclear reactors by applying it to suitably adjusted (through the addition of polyvinyl alcohol) for viscosity simulant systems.

This is a carefully performed experimental work of a potentially useful micromixing and droplet formation device that convincingly demonstrates its performance in generating uniform spherical droplets of well-mixed aqueous ingredients in silicon oil in throughputs of the order of mL/min. However, there are a few concerns regarding the window of applications that can covered, as described in the detailed comments below, that the authors are asked to respond with in a revised version before their work can be recommended for publication. We thank the reviewers for acknowledging the strong performance of this work and the quality of the presentation. We address the comments as follows:

(1) The size of the produced droplets is marginally in the micron scale (more in the millimeter scale)---see, for example, Figs. 5, 6, 8. Is this a restriction of the device?  What is the window of feasible sizes that can be realistically produced (by the tested or suitably scaled device) and what are the implications to the obtained throughputs?

We thank the Reviewer for the careful consideration. This is indeed a restriction of this device, because the inner diameter of the outlet pipe of the T-junction micromixer in this device is 2.65 mm, which determines the upper limit of the droplet size produced. It has been proven in the previous research of the research group [1]. When droplets are generated in a small pipe such as 1~1.6mm at the flow rate of mL/min, the flow resistance in the pipe will be very large, resulting in slow movement of the generated droplets and the syringe pumps will stop working due to excessive pressure. Therefore, we set the inner diameter of the outlet pipe to 2.65 mm to reduce the flow resistance, so as to achieve a high-throughput instant mixing of mL/min. In the introduction of the manuscript, it is found that the flow condition is rarely achieved on other researchers’ micromixers yet. The system can produce droplets with a size of 100~3000 μm, by changing the size of the outlet pipe, the flow rate of the continuous phase and the dispersed phase, and the viscosity of the continuous phase, etc. If 100 μm size droplets need to be generated, the size of the outlet pipe and the flow rate of the dispersed phase need to be reduced, and the flow rate of the continuous phase needs to be increased appropriately. This means that the flow rate of the dispersed phase may be μL/min, which greatly reduces the throughput of droplets. When the system is operated under the best conditions in the manuscript, the flow rate of the dispersed phase is 0.5 mL/min. At this time, the size of the droplet produced is 1.118 mm. According to the volume formula of the sphere, the volume of a single droplet can be calculated as:

Therefore, under optimal conditions, the system can produce approximately 683 droplets per minute.

(2) A significant motivation for this work is its potential use for the preparation of UO2 and ZrO2 gel microspheres. However, the authors have not applied their device to the actual systems, but rather to so called “simulants” though which they simply matched the viscosities of the actual suspensions.  As those suspensions may well have a much more complicated inner structure that that of the simulants, a concern is whether the latter can be used to actually predict their mixing behavior.  The density differences are also notable.  Is there additional evidence pointing out for the suitability of the proposed simulants?  Also, in the previously quoted works (namely, reference 28 provided, as well as previous work by some of the authors that has not be mentioned here—see reference 1 below) the droplets size was much smaller of the order of 100 microns.  Is that a factor?  Related to remark (1) above, can the proposed device be adjusted to the smaller size and how is that expected to affect the throughput?  Finally, reference 1 below, suggests that external gelation leads to better quality particles as opposed to the internal gelation that is attempted to be simulated here.  Can the deficiencies detected in reference 1 associated with internal gelation production be circumvented?  A more direct comparison of the proposed device against that used in reference 1 is recommended.

Special thanks to the Reviewer for your good and careful comments. As Reviewer suggested that the viscosity and surface tension of the simulated solution are similar to the actual solution, but its density is somewhat different from the actual solution. However, in microfluidics, surface tension is the main factor and gravity can be ignored. This is also one of the basic characteristics of microfluidics. Therefore, the density difference between the simulated solution and the actual solution can be ignored. It is really true as Reviewer suggested that the actual solution is more complicated than the simulated solution and may involve chemical reactions during the mixing process. We will use this system to actually prepare zirconia microspheres in subsequent studies to prove the feasibility of the system.

Thanks again to the reviewers for their careful check. References 1 and 28 are the previous research of the group. Reference 1 uses the external gelation process and Reference 28 uses the internal gelation process. In Reference 1, small-sized cerium-stabilized zirconia microspheres were prepared to increase the packing density of particles. In this manuscript, the system aims to mix the solution into an unstable internal gelation broth, and then use silicone oil to cut the broth into droplets, so as to achieve room temperature preparation of zirconia or uranium oxide microspheres. Generating 100 μm droplets is not the main purpose of the system, and it will cause the system to produce great flow resistance, which is not conducive to the realization of high throughput.

In Reference 1, the cerium-stabilized zirconia microspheres prepared by the external gelation process have higher crush strength and density than those prepared by the internal gelation process. It can be seen from Fig.S1 that the external gelation process involves the reaction of metal ions with external ammonia water to obtain gel microspheres. The internal gelation process uses metal ions to react with the ammonia decomposed by hexamethylenetetramine inside the broth droplets to obtain gel microspheres. Essentially, the external gelation process is a mass transfer solidifying process, while the internal gelation process is a heat transfer solidifying process. The solidified shell layer during the gelation process will hinder the mass transfer of ammonia without hindering the transfer of temperature [2]. Therefore, compared with the external gelation process, the microspheres prepared by the internal gelation process have better sphericity. This result is also confirmed in Reference 1. However, the internal gelation process has the difficulties of slightly uneven size of the prepared microspheres and the need to cool the broth. Therefore, a novel high-throughput and instant-mixing microfluidic system is designed in this manuscript to solve the difficulties of the internal gelation process and realize the preparation of monodisperse oxide microspheres at room temperature without cooling, instead of comparing with the external gelation process.

Fig.S1 | Schematic diagram of sol-gel method. A, the external gelation process.

B, the internal gelation process.

References

[1] Guogao Wang, Jingtao Ma, Yong Gao, Xingyu Zhao, Shaochang Hao, Changsheng Deng, Bing Liu, A comparative study of small-size ceria–zirconia microspheres fabricated by external and internal gelation, J Sol-Gel Sci Technol (2016) 78:673–681.

[2] Liang S , Li J , Li X , et al. Microfluidic fabrication of ceramic microspheres with controlled morphologies[J]. Journal of the American Ceramic Society, 2018, 101(9).

[28]. Gao Y, Ma J, Zhao X, Hao S, Deng C, Liu B, Franks G (2015) An Improved Internal Gelation Process for Preparing ZrO2 Ceramic Microspheres Without Cooling the Precursor Solution. Journal of the American Ceramic Society 98 (9):2732-2737.

(3) Some editorial remarks:

  1. a) Line 101, it is the first-time polyvinyl alcohol is mentioned, but not the acronym (PVA) that is though mentioned several times later---this needs to be added.
  2. b) In figure 3A, B and C along with the larger droplets there are much smaller ones: those are bubbles?
  3. c) Line 184: It is mentioned that the performance slows down, however, Figure 4 shows that rather it reaches a plateau.
  4. d) Line 196: The addition of surfactant is mentioned, but not what the surfactant was and the effects that its addition to the surface tension are---those are needed to better interpret the results shown in Figure 5.
  5. e) Line 466, reference 27: Please correct the name of the journal.

Thanks for the helpful comments. Considering the Reviewer’s suggestion, we have made correction according to the Reviewer’s comments in the corresponding position in the manuscript. When the residual gas in the system has been discharged and the droplets are stably produced, the droplets will be collected and photographed. Therefore, the small droplets in Figures 3A, 3B, and 3C may be satellite droplets produced by the system.

We are very sorry for our incorrect writing. The words “slow down” is corrected to “reaches plateau” and the explanation of surfactants is added. We are very sorry for our negligence of the Reference 27 of the name of the journal. Special thanks to the Reviewer for your good and careful comments again.

(Item #1, #2, #3 and #4 in the outline)

Reviewer 2 Report

In their manuscript, the authors present a microfluidic system to mix reagents in a fast manner and to generate micro droplets downstream. In comparison to other micromixers on the market, they claim to achieve better mixing performance and throughput.

The system is highly characterized, and the application demonstrated using model reagents (color pigments). The characterization is well done. The figures are understandable and clean.

When reading the manuscript, it is not clear from the beginning why  something should clog the system and why they emphasize the easy cleaning. The purpose of the system is not well explained at the beginning (in the introduction). Some more text on what it is about that they want to achieve at the end and why it is challenging would be helpful.

The technological level of the system is not very high, but it serves the purpose. It is a very simple system and can easily be built. It would be nice to see some more construction plans or images as supplementary material.

I think the work can help others in the field to build up microfluidic systems, where fast in-line mixing is important and consequent droplet generation is needed. Therefore, with some additional explanation for clarity I recommend publishing the work.

Author Response

5.Introduction, paragraph 2 (Line 4~10 on the third page), we add the reason about the clog of micromixers and explain why micromixers need cleaning.

6.Introduction, paragraph 2 (Line 11~12 on the third page), we add the challenge of the micromixer and supplement the necessity of the reusable, low-cost, high-throughput micromixer.

7.Introduction, paragraph 3 (Line 16~18 on the third page), we add the purpose and some features of the system.

Replies to the comments of Reviewer 2:

In their manuscript, the authors present a microfluidic system to mix reagents in a fast manner and to generate micro droplets downstream. In comparison to other micromixers on the market, they claim to achieve better mixing performance and throughput. The system is highly characterized, and the application demonstrated using model reagents (color pigments). The characterization is well done. The figures are understandable and clean. We acknowledge the reviewer for the positive comments. Clarification to the comments are given below:

(1) When reading the manuscript, it is not clear from the beginning why something should clog the system and why they emphasize the easy cleaning. The purpose of the system is not well explained at the beginning (in the introduction). Some more text on what it is about that they want to achieve at the end and why it is challenging would be helpful.

We have made correction according to the Reviewer’s comments. As in the introduction, the researchers used a micromixer to prepare BaSO4 or Fe3O4 micro-nano particles. The principle is that the two solutions rapidly generate insoluble substances in the micromixer. When the mixing is uneven or the mixing time is too long, a large amount of insoluble matter will stay in the microchannel and block the microchannel. Micromixers are generally packaged by thermal bonding. When the microchannel in the micromixer is blocked, the micromixer is generally scrapped. Therefore, the system is disassembling and easy to clean the insoluble matter in the microchannels, realizing the reuse of the system and greatly reducing the cost. By summarizing some problems of previous micromixers, the reusable, low-cost, high-throughput instant mixing system is designed, which is expected to be used for the mixing of solutions and the preparation of micro-nano ceramic particles. Special thanks to you for your good comments.

(Item #5, #6 and #7 in the outline)

(2) The technological level of the system is not very high, but it serves the purpose. It is a very simple system and can easily be built. It would be nice to see some more construction plans or images as supplementary material.

It is really true as Reviewer suggested that the technology of the system is simple, but it can efficiently achieve high-throughput and instant mixing at a flow rate of mL/min and the system can be disassembled to clean the clogged part of the microchannel. In comparison with the mixing performance of mainstream micromixers on the market, the system has the best mixing performance. However, the complex microchannel structure in the Serpentine micromixing chip in Fig.7C in the manuscript has poor mixing performance. This is because the microchannel structure of the system is simple, making its flow resistance relatively low, making it easier to achieve high-throughput instant mixing to prepare uniformly colored, monodispersed droplets. Of course, because the system adopts a detachable design, when the flow rate of the microchannel reaches mL/min, the internal pressure of the entire system will be very high, so special attention should be paid to the leakage of the liquid at the pipe connection, which may cause uneven mixing. This is also one of the difficult technologies of the system. The physical map of the system is shown in Fig.S2 of the supplementary material. Two solutions of different colors are quickly mixed into a solution of another color by the magnetic micromixer, and the mixed solution is then cut into droplets of uniform size by the silicone oil in the T-junction microfluidic device. Thanks again for your valuable comment.

Fig.S2. A, physical image of the high-throughput and instant mixing droplet microfluidic system.

B, the physical image of the magneton micromixer.

(3) I think the work can help others in the field to build up microfluidic systems, where fast in-line mixing is important and consequent droplet generation is needed.

We gratefully appreciate for Reviewer valuable suggestion. In fact, as mentioned in the introduction, mixing is a necessary step in a chemical reaction. The system can produce droplets of uniform size at a flow rate of mL/min. This is expected to be applied to some dangerous chemical reactions and the preparation of micro-nano ceramic microspheres, such as the preparation of zirconia, silicon oxide, and titanium oxide microspheres. Moreover, the structure of the system is relatively simple, which is conducive to the popularization of the system’s technology.

Reviewer 3 Report

There are some weaknesses through the manuscript which need improvement. Therefore, the submitted manuscript cannot be accepted for publication in this form, but it has a chance of acceptance after a major revision. My comments and suggestions are as follows:

1- Abstract gives information on the main feature of the performed study, but some details about the proposed microfluidic system must be added. The current abstract is too short.

2- Authors must clarify necessity of the performed research. Objectives of the study, must be clearly mentioned in the last part of introduction.

3- The literature study must be enriched. In this respect, authors must read and refer to the relevant papers: (a) https://doi.org/10.1007/s40094-016-0217-9 and (b) https://arxiv.org/abs/1910.00002 and (c) https://doi.org/10.1016/j.seppur.2019.01.010

4- As the manuscript deals with experiments (construction of microfluidic system), authors must illustrate some figures and show the fabrication process and produced system.

5- It is necessary to present reference for values mentioned in Table 1.

6- The main reference of each formula must be cited. Also, all parameters must be introduced.

7- Standard deviation in presented results must be discussed.

8- In its language layer, the manuscript should be considered for English language editing. There are sentences which have to be rewritten.

9- The conclusion is too short! It must be more than just a summary of the manuscript. List of references must be updated based on the proposed papers. Please provide all changes by red color in the revised version.

Author Response

Replies to the comments of Reviewer 3:

There are some weaknesses through the manuscript which need improvement. Therefore, the submitted manuscript cannot be accepted for publication in this form, but it has a chance of acceptance after a major revision. We have studied comments carefully and have made correction which we hope meet with approval. Revised portion are marked in red in the paper. The main corrections in the paper and the responds to the reviewer’s comments are as flowing:

(1) Abstract gives information on the main feature of the performed study, but some details about the proposed microfluidic system must be added. The current abstract is too short.

We are very sorry for our incorrect writing. Considering the Reviewer’s suggestion, we have corrected the abstract.

According to the references sent by the Reviewer, the abstract is rewritten in accordance with the purpose, method, and conclusion of the paper.

(Item #8 in the outline)

(2) Authors must clarify necessity of the performed research. Objectives of the study, must be clearly mentioned in the last part of introduction.

Thanks for the Reviewer’s good suggestion. In the second paragraph of the introduction, we supplement the necessity of research. And in the third paragraph of the introduction, we supplement the purpose of the system.

(Items #6 and #7 in the outline)

(3) The literature study must be enriched. In this respect, authors must read and refer to the relevant papers: (a) https://doi.org/10.1007/s40094-016-0217-9 and (b) https://arxiv.org/abs/1910.00002 and (c) https://doi.org/10.1016/j.seppur.2019.01.010.

Thanks for the good suggestion. After reading these three literatures in detail, the numerical simulation and calculation capabilities of these literatures have benefited us a lot. In Section 3.5, we add the three new literatures and update references.

(Items #9 and #14 in the outline)

(4) As the manuscript deals with experiments (construction of microfluidic system), authors must illustrate some figures and show the fabrication process and produced system.

It is really true as Reviewer suggested that some figures about the microfluidic system need to be listed. In this regard, we will upload supplementary materials on microfluidic systems, as shown in Fig.S2. Two solutions of different colors are quickly mixed into a solution of another color by the magnetic micromixer, and the mixed solution is then cut into droplets of uniform size by the silicone oil in the T-junction microfluidic device. Thanks again for your valuable comment.

(5) It is necessary to present reference for values mentioned in Table 1.

Thanks for Reviewer’s feedback and suggestions. We add the references to the density, viscosity and surface tension of zirconium broth and uranium broth.

(Items #10 in the outline)

(6) The main reference of each formula must be cited. Also, all parameters must be introduced.

Thanks for the careful and good suggestion. In Section 2.3 (Line 19~20 on the seventh page), we add the explanation of “n” parameter in Equation (1). In Section 3.7 (Line 15 on the eighteenth page), we add the explanation of “mn+” parameter in Equation (2).

(Items #11 and #12 in the outline)

(7) Standard deviation in presented results must be discussed.

Thank you very much for the reviewer's comments. The formula for standard deviation (error bars) is as follows:

Where  is the sample value and  is the average value of the sample and N is the number of samples.

The coefficient of variation (CV) is actually the ratio of the standard deviation to the mean:

Therefore, MP actually includes the concept of standard deviation, making MP comparisons without error bars. Similarly, in the comparison of the coefficient of variation of droplet size, there is no error bar. Regarding the influence of different factors on the droplet sizes, the droplet sizes in the picture of manuscript all have error bars. Therefore, the data in the manuscript all considers the standard deviation.

(8) In its language layer, the manuscript should be considered for English language editing. There are sentences which have to be rewritten.

Thank you very much for the suggestions of the reviewer. The manuscript has been polished by a native English speaker. Thanks again for your correction.

(9) The conclusion is too short! It must be more than just a summary of the manuscript. List of references must be updated based on the proposed papers.

We are very sorry for our incorrect writing. We have made correction according to the Reviewer’s comments.

(Items #13 in the outline)

We would like to thank the reviewers for their careful readings and valuable comments again. We believe the constructive feedback will improve the paper and increase its potential impact to the community. Looking forward to hearing from you. Thank you and best regards.

Jingtao MA (Tsinghua) (On behalf of all authors)

Round 2

Reviewer 3 Report

The paper has been improved and corresponding modifications have been conducted. Please check the reference list to write correct name of authors  (e.g., Ref. 27 in revised version).

In my opinion, the current version can be considered for publication.